Visualization of oxygen distribution patterns caused by coral and algae

Haas Andreas F. 1 2 andreas.florian.haas@gmail.com
Gregg Allison K. 1
Smith Jennifer E. 2
Abieri Maria L. 1 3
Hatay Mark 1
Rohwer Forest 1
1 Department of Biology, San Diego State University , United States
2 Scripps Institution of Oceanography, University of California , San Diego , United States
3 Institute of Biology, Department of Marine Biology, Federal University of Rio de Janeiro , Brazil
Thompson Fabiano
Electronic publication date: 2013 Jul 16
Publication date: 2013
Volume: 1
Electronic Location ID: e106
Received 2013 Apr 30; Accepted 2013 Jun 25
Copyright: © 2013 Haas et al.
Copyright year: 2013
Copyright holder: Haas et al.
License: This is an open access article distributed under the terms of the Creative Commons Attribution License, which permits unrestricted use, distribution, and reproduction in any medium, provided the original author and source are credited.
License URL: https://creativecommons.org/licenses/by/3.0/

Keywords: Planar optode, Dissolved oxygen, Interaction, Algae, Coral, Two dimensional visualization

Funding: NSF Grant Nos. OCE-0927448 OCE-0927415 DEB-1046413 Funding for this study was provided by the National Science Foundation to Principal Investigators JES (NSF Grant No. OCE-0927448) and FR (NSF Grant No. OCE-0927448 OCE-0927415 and DEB-1046413). Additional funding was provided by CAPES-FIPSE and Rede Abrolhos (SISBIOTA/CAPES, CNPq, FAPES), Brazil. The funders had no role in study design, data collection and analysis, decision to publish, or preparation of the manuscript.

==============================
Planar optodes were used to visualize oxygen distribution patterns associated with a coral reef associated green algae (Chaetomorpha sp.) and a hermatypic coral (Favia sp.) separately, as standalone organisms, and placed in close proximity mimicking coral-algal interactions. Oxygen patterns were assessed in light and dark conditions and under varying flow regimes. The images show discrete high oxygen concentration regions above the organisms during lighted periods and low oxygen in the dark. Size and orientation of these areas were dependent on flow regime. For corals and algae in close proximity the 2D optodes show areas of extremely low oxygen concentration at the interaction interfaces under both dark (18.4 ± 7.7 µmol O2 L- 1) and daylight (97.9 ± 27.5 µmol O2 L- 1) conditions. These images present the first two-dimensional visualization of oxygen gradients generated by benthic reef algae and corals under varying flow conditions and provide a 2D depiction of previously observed hypoxic zones at coral algae interfaces. This approach allows for visualization of locally confined, distinctive alterations of oxygen concentrations facilitated by benthic organisms and provides compelling evidence for hypoxic conditions at coral-algae interaction zones.

Introduction

Reef building corals and macroalgae can both act as ecosystem engineers by physically and chemically altering their environment (Jones, Lawton & Shachak, 1994; Wild et al., 2004) but they do so in different ways (Wild et al., 2010; Haas et al., 2011). One of the key water column parameters affected by both of these organismal groups (and many others) is oxygen availability (Wild et al., 2010; Niggl, Haas & Wild, 2010). Dissolved oxygen concentrations in the reef water column can vary by more than 50% between daylight hours with active benthic photosynthesis and night, where respiration by the reef community results in net oxygen consumption (Niggl, Haas & Wild, 2010; Haas et al., 2010).

Many studies have investigated variation in oxygen concentrations in different marine environments and on different scales. While some studies have targeted small scale oxygen changes in boundary layers (de Beer et al., 1994; Larkum, Koch & Kühl, 2003), coral interstices (Shashar, Cohen & Loya, 1993; Kühl et al., 1995), and coral-algal interaction interfaces (Smith et al., 2006; Barott et al., 2009), others have examined whole reef areas (Middleton et al., 1994; Niggl, Haas & Wild, 2010) or even influences of warming on dissolved oxygen concentrations in oceans around the world (Stramma et al., 2010). All of these studies assessed changes in oxygen concentrations by point measurements with varying spatial and temporal resolution across a gradient or grid. Water column oxygen concentrations are clearly not static and are going to vary across space and over time due to biological processes and hydrodynamics (Rasheed et al., 2004). This variability makes it difficult to understand how individual organisms or communities affect or are affected by oxygen concentrations on coral reefs.

Planar oxygen optodes have been used to resolve oxygen dynamics in a variety of complex benthic environments such as animal burrows (Volkenborn et al., 2010), marine plant root systems (Frederiksen & Glud, 2006), permeable sands (Polerecky et al., 2005), and even in endolithic algal communities within coral holobionts (Kühl et al., 2008). These optodes function via a luminescent indicator that is differentially quenched in the presence of oxygen, which allows for the two-dimensional visualization of oxygen distribution patterns (Holst et al., 1998; Oguri, Kitazato & Glud, 2006). However, planar optodes have never been used to investigate oxygen distribution patterns in the water column, and until now have not been used for in situ measurements. In the present study we used a planar optode system (Larsen et al., 2011), modified for in situ applications (AK Gregg, unpublished data), to visualize the two-dimensional oxygen concentrations patterns generated by coral reef associated macroalgae and corals separately and in close proximity to one another. This allows for the assessment of the influence of the respective organisms on DO concentration, and the potential variation resulting from competition/interaction processes on oxygen distribution patterns in the surrounding water column.

Material and Methods

Experimental setup

Oxygen distribution patterns facilitated by independent coral and algal samples and samples of each taxon placed in close proximity to one another (mimicking coral-algae interactions), were assessed under flow and no-flow conditions. To generate unidirectional flow a flow chamber was constructed from clear acrylic with inner dimensions of 16 × 16 × 50 cm. The flow chamber was constantly supplied with filtered (nominal pore size 50 µm) and temperature controlled (26.3 ± 0.4°C) seawater. Flow rates were adjusted to the desired flow regime by controlling the water supply to the chamber. Flow rates were calculated by determining the water quantity passing through the chamber and visually verified by tracking the passage of neutrally buoyant particles through the chamber. Temperature was recorded every minute by an Onset HOBO® Pendant UA-002-64 temperature logger throughout the duration of each experiment.

Algae (Chaetomorpha sp.) and coral (Favia sp.) specimens used in this experiment were provided by the Birch Aquarium at Scripps Institution of Oceanography. Corals were mounted with coral cement (Instant Ocean, Holdfast® Epoxy Stick) and algae were fixed with zip ties to identical ceramic tiles (5.0 × 5.0 × 0.5 cm) that fit exactly in a cut out square on the bottom of the flow chamber to prevent turbulence generated by the ceramic tile base. Specimens were acclimated for at least 48 h in a cultivation tank with temperature controlled (Temp = 26.1 ± 0.5°C) flow through seawater to recover from handling as described above.

For each experimental run (n = 3 for each “run”) the tile-attached organisms were placed in the designated notch of the flow chamber and an optode sheet (16 × 16 × 30 cm) was placed in a vertical plane over them (Fig. 1A). All samples were exposed to four different treatments, comprising artificial daylight and dark conditions in a non-moving water body and under flow rates of ∼5 cm s- 1 at distances of >5 cm from chamber walls. Artificial light was provided by 2 × 54 W 6000 K Aquablue + , 1 × 54 W 6000 K Midday, and 1 × 54 W Actinic+ aquarium lights (Geismann, Germany), mounted 80 cm above the experimental chambers resulting in photosynthetic active radiation of 160 µmol quanta m- 2 s-1 as measured by a LI-COR LI-193 Spherical Quantum Sensor. Samples were incubated for at least 3 h in the respective experimental conditions and pictures of the planar optodes were captured every 30 min. To verify oxygen concentrations patterns visualized by the optodes, 3 hand-held LBOD101 luminescent oxygen meters (Hach Lange, Germany, precision 0.01 mg l-1, accuracy ± 0.05%) were mounted in control setups, in- and outside of the expected oxygen plumes as reference (deviation was always <5%).

Figure 1 Experimental setup.

(A) Flow chamber: (i) Canon D10 camera equipped with Schott 530 nm long pass filter (ii) 445 nm LED with a 470 nm Blue Dichroic short pass filter (iii) Planar oxygen optode, mounted in vertical plane over (iv) benthic organism(s) (v) Diffuser to create consistent laminar water flow. (B) Example of a picture taken to visualize oxygen distribution generated by algae and coral in close proximity, subjected to flow conditions (indicated by arrows) during dark incubation (indicated by moon). Oxygen scale represents approximate values. (C) Plot of dissolved oxygen concentrations at 26.0°C as seen in Fig. 1B; analyzed in MATLAB. Oxygen scale represents exact values. Grayscale represents coral and algae specimens; not representative of oxygen concentration.

Planar optode images

Camera and light settings – Images of planar optodes (Fig. 1B) were taken visualize oxygen concentrations in two-dimensions along the optode sheet (modified from Larsen et al., 2011). Planar oxygen optodes were photographed using a G11 (Canon, USA) placed at a distance of ∼25 cm from the flow chamber; images were captured in RAW format. All photographs were taken with identical settings of ISO 200, f∖8 and shutter speed of 1.3 s.

Four Rebel Royal Blue light emitting diodes (LED) with a λ-peak of 445 nm (Phillips-Luxeon, Canada) were used as the excitation source in combination with a 470 nm short pass filter (UQG Optics, UK). To prevent the excitation source from contaminating the luminescent signal, a Schott 530 nm long pass filter (UQG Optics, UK) was mounted on the camera lens. All images were taken in the absence of ambient light.

Oxygen planar optodes

The oxygen sensitive optical indicator platinum (II) octaethylporphyrin (PtOEP) was used in combination with the coumarin antenna dye macrolex® fluorescence yellow 10GN (MY) (for details see Larsen et al., 2011). The luminescence of PtOEP, with a peak wavelength of 650 nm, is quenched in the presence of oxygen and its intensity therefore dependent on the oxygen concentration, whereas the MY emission intensity, with a peak wavelength of 515 nm, remains unaffected by oxygen concentrations and is therefore constant. In this system, only part of the excitation energy collected by the antenna is transferred to the indicator dye (Mayr et al., 2009) and the remaining energy is emitted from the antenna dye itself, thereby acting as an internal reference (Larsen et al., 2011). Absolute oxygen concentrations can therefore be calculated using the ratio of red pixel intensity (oxygen-dependent dye) to green pixel intensity (oxygen-independent dye). Optode sheets were prepared using 0.04% (w/v) of both PtOEP and MY, along with 4% (wt/vol) polystyrene, dissolved in chloroform (Larsen et al., 2011), and spread onto 0.125 mm thick PET film (GoodfellowUSA, USA). The final thickness of the optode sheet was approximately 10 microns. The sheets were then cut to fit in the vertical plane of the flow chamber, leaving a 6 cm wide semicircle cut-out for the biological samples.

Image analysis

The RAW images were imported into the image-processing program RawHide (v0.88.001, My-Spot Software, USA). Resulting pixel size for TIFF images was 3.1 megapixels. Each RAW file was converted into three 16-bit TIFF images (i.e. red, averaged green and blue color channel). The pixel information from the red and green channel images were imported into MATLAB and further analyzed using the image toolbox (Fig. 1C, Supplemental Information 1). The red and green intensity values (0–65,536 for a 16-bit image) were obtained for each pixel and used to calculate the pixel intensity ration (R) (Eq. (1)). The resulting ratios were used in the modified Stern-Volmer equation (Klimant, Meyer & Kühl, 1995), where α is the unquenched portion of the indicator, R0 is the ratio at anoxia, C is the concentration of oxygen and Ksv is the Stern-Volmer constant. (1) R=intensity of red−intensity of greenintensity of green

(2) RR0=α+(1−α)11+Ksv⋅C

To establish the constants necessary to calculate absolute oxygen concentrations, a calibration of each optode sheet under identical temperature conditions was performed prior to the experiment. Each optode sheet was calibrated by taking images at known oxygen concentrations. Filtered and temperature controlled seawater was doused with nitrogen gas to obtain 8 different concentrations of oxygen ranging from 100% air-saturation to anoxia. Starting concentrations were approximately 250 µM oxygen and an image was taken at approximately every 30 µM step until the seawater was anoxic. The resulting oxygen concentrations were constantly measured for comparison using an LBOD101 luminescent oxygen probe. From this calibration, we determined our values for α and Ksv for each respective optode (Eq. (2), example given in Fig. S1) using a non-linear regression (Prism version 5, R version 2.13.2). Regression curves are included in Fig. S1. If not further specified, values are given as mean ± standard error (SE).

Results and Discussion

Here we visualized two-dimensional oxygen distribution patterns generated by distinct functional groups of coral reef associated ecosystem engineers using planar optodes. Analyses of the images revealed discrete regions of oxygen concentration changes in their surrounding water column. Oxygen concentrations in the surrounding water columns were comparable between light (161.8 ± 10.4 µmol O2 L- 1) and dark incubations (164.5 ± 13.5 µmol O2 L- 1). Measurements showed that during daylight hours, and with no flow, oxygen concentrations were elevated by 91–210 µmol O2 L- 1 in the water overlying the algae (Fig. 2A) and 19–149 µmol O2 L- 1 above the investigated coral (Fig. 2B). Maximum oxygen concentrations (∼400 µmol O2 L- 1) were considerably above (∼200% oxygen saturation) the seawater saturation limit in these oxygen plumes. Oxygen concentration visualization in the non-flow water conditions showed minimal lateral influences of the biological samples on the water column next to them. The released oxygen rapidly rose to the surface, suggesting that under low flow conditions (e.g., ebb tide) a major part of the oxygen produced, may not necessarily be available to the surrounding benthic community but is rather shunted vertically through the water column. This visible phenomenon was more pronounced in the water column overlying algae than that of the corals (oxygen concentration difference between inside and outside the plumes for algae: 148.3 ± 24.7 µmol O2 L- 1 and coral: 66.7 ± 30.7 µmol O2 L- 1 standalone treatments).

Figure 2 Oxygen patterns over coral and algae during light and dark.

Examples of pictures taken of planar optodes mounted over algae (A, C) and corals (B, D) subjected to no-flow conditions during artificial daylight (A, B) and dark (C, D) incubations (indicated by sun and moon icons). Note the plumes of oxygen (green signal) rising from the organisms. Oxygen scale represents approximate values.

Organismal respiration generated reverse patterns of oxygen distribution during dark conditions. Concentration differences were in the same range (77–107 µmol O2 L- 1 for corals and 34–160 µmol O2 L- 1 for algae), but spatially more confined and with less pronounced vertical patterns (Figs. 2C and 2D). Minimum oxygen concentrations of 79.8 ± 16.4 µmol O2 L- 1 for algae and 70.4 ± 13.1 µmol O2 L- 1 for corals were detectable within 5–10 mm of the organisms. There were no noticeable differences in the distribution patterns of low oxygen zones facilitated by standalone coral and benthic algae during dark conditions.

Water flow of ∼5 cm s- 1, simulating flow regimes in ranges which can be found in situ in back reef environments (Hench, Leichter & Monismith, 2008), decreased intensity and changed the direction of the plumes (Fig. 3). Increased (Fig. 3A) or decreased (Fig. 3B) oxygen was observed downstream of all specimens under light and dark conditions, respectively. Relatively sharp boundaries in oxygen concentrations were maintained over distances in the range of centimeters (Figs. 3C and 3D). Previous studies have suggested that oxygen distribution patterns facilitated by benthic organisms are determined by light conditions and the boundary layer thickness, and as a function of flow and surface type (Brown, 2012). Additionally Marhaver et al. (2013) described current dependant distribution patterns of organism-associated microbes in ranges of up to 1 m around coral colonies. These findings collectively suggest that benthic organisms living adjacent to one another may experience large fluctuations in key parameters such as oxygen availability, pH values (Smith et al., 2013), microbial community structure, and this variability will be dependent on irradiance, hydrodynamics and their surrounding cohabitants.

Figure 3 Algae generated oxygen patterns under flow conditions.

Photographic visualization of 2-dimensional oxygen distribution patterns around algae in under flow conditions during daylight (A) and dark (B) conditions (indicated by sun and moon icons). Arrows indicate direction of water movement. (C, D) MATLAB processed Plot of dissolved oxygen concentrations at 26.0°C as seen in corresponding pictures A and B. Oxygen scale represents exact values. Grayscale represents coral and algae specimens; not representative of oxygen concentration.

Although hydrodynamic conditions might have been affected by wall effects along the planar optode, our study reveals that benthic organisms can drive distinctive, locally heterogeneous patterns in oxygen concentration in their surroundings. The images generated in this study suggest that single point oxygen concentration measurements, that are often made to quantify variability across space and time, may not accurately represent the true oxygen dynamics. Here we found that oxygen concentrations may differ by more than 60% along distances of less than 10 mm. Recent efforts in developing in situ applicable oxygen sensitive planar optodes (Glud et al., 2001, AK Gregg, unpublished data) could however provide a useful tool to allow for a better understanding of oxygen fluxes in the 2 or even 3-dimensional space of a highly complex coral reef environment.

The images generated from planar optodes on oxygen dynamics of the interactions between coral and algae when placed in close proximity to one another yielded comparable patterns where there was high oxygen production during the day and consumption at night on the sample surfaces not at the zone of interaction. However, highly reduced oxygen concentrations were present in the immediate proximity of the interfaces. During dark incubations these interaction zones showed the lowest oxygen concentrations measured (18.4 ± 7.7 µmol O2 L- 1, Fig. 4B). Even during light treatments with noticeable oxygen generation of both organisms on the distal side, the interfaces were engulfed in spatially confined, low oxygen zones (97.9 ± 27.5 µmol O2 L- 1, Fig. 4A). Although these zones of decreased oxygen concentrations were most pronounced in environments with no induced water movement, they could also be detected under flow conditions and in situ, in a pilot study to develop new in situ Submersible Oxygen Optode Recording (SOOpR) system, by AK Gregg (unpublished data).

Figure 4 Oxygen patterns generated by coral algae interaction.

Examples of pictures taken of planar optodes mounted over corals and algae in close proximity mimicking coral algae interaction processes during daylight (A) and dark (B) conditions. Note the low oxygen concentration zones at the interfaces of coral algal interactions visible even during light conditions permitting photosynthetic oxygen production. Oxygen scale represents approximate values.

These hypoxic, or even anoxic zones at interaction interfaces, have been suggested to drive coral-algal competition processes (Smith et al., 2006), and have been noted by other studies (Barott et al., 2009; Wangpraseurt et al., 2012), but could never be shown in their full extent and under varying light conditions and hydrodynamics. The mechanisms underlying these patterns are currently subject to discussion. While some studies attribute these anoxic conditions to modifications in small scale topography, with no immediate effects on interaction processes, others suggest increased microbial oxygen demand as the underlying cause. Previous studies have shown that phase shifts from slow-growing reef-building organisms to fleshy algae, as a result of increased anthropogenic influences (Francini-Filho et al., 2013), were accompanied by shifts from photosynthetic microbial communities to higher abundances of archaeal and viral sequences and more bacterial pathogens (Bruce et al., 2012). These shifts may also result in shifts in microbial metabolic rates at coral algal interaction zones.

The DDAM (dissolved organic matter, disease, algae, microbes) model introduced by Dinsdale & Rohwer (2011) hypothesizes that algae release bioavailable organic compounds (Haas et al., 2011), which then facilitate microbial growth and respiration (Wild et al., 2010; Nelson et al., 2013), particularly of opportunistic pathogens (Dinsdale et al., 2008). This likely leads to higher morbidity and mortality of corals (Kuntz et al., 2005; Kline et al., 2006) as a consequence of both increases in coral pathogens and dramatically reduced oxygen availability (reviewed in Barott & Rohwer, 2012).

The data presented here provides compelling, visible evidence of the existence of highly variable patterns in oxygen distribution patterns associated with benthic reef organisms. Despite this variability we see clear patterns of oxygen decline associated with the interface between interacting coral and algae during both dark and daylight conditions. This implies that the driver for these pronounced low oxygen regions is not solely an effect of low metabolic activity facilitated by a depression in the local topography (Wangpraseurt et al., 2012), but can more likely be attributed to (a) increases in heterotrophic metabolism of the competitors as a general response to stress (Moberg et al., 1997; Abrego et al., 2008), or (b) increases in microbial oxygen demand (Smith et al., 2006; Barott & Rohwer, 2012). This new technique thus provides an opportunity to clearly visualize and quantify these patterns. Further our results suggest that the pattern of algae induced hypoxia in competitive interactions between corals and algae may be a common phenomenon.

Supplemental Information

Supplemental Information 1 Matlab script

Matlab script to generate colormap images of spatial oxygen dynamics.

Click here for additional data file.

Figure S1 Example of optode calibration curve

Optode calibration was done with a non-linear regression model using the one site Stern-Volmer equation (Eq. (2)). Stern-Volmer constant (Ksv = 6.1 × 10−3 µM−1) and unquenched portion of the indicator (1.27 × 10−9) for the here used optode sheet was derived from this regression.

Click here for additional data file.

We especially thank Morten Larsen and Ronnie N. Glud for their assistance with the planar optode preparation and camera system set up. Further we thank Peter Salamon, Jim Nulton, and the Undergraduate Biomath Program for their help with the picture analysis (funded by NSF Grant No. 0827278). We also thank Franklin Holub for his support with Rawhide. Thanks to Birch Aquarium at Scripps for providing the algal specimens.

Additional Information and Declarations

Competing Interests

Author Contributions

The authors declare no competing interests.

Andreas F. Haas and Allison K. Gregg conceived and designed the experiments, performed the experiments, analyzed the data, wrote the paper.

Jennifer E. Smith and Forest Rohwer conceived and designed the experiments, analyzed the data, contributed reagents/materials/analysis tools, wrote the paper.

Maria L. Abieri performed the experiments, analyzed the data, wrote the paper.

Mark Hatay conceived and designed the experiments, performed the experiments, analyzed the data, contributed reagents/materials/analysis tools, wrote the paper.

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
