# Peer review of "Visualization of oxygen distribution patterns caused by coral and algae"

_PeerJ, doi:10.7717/peerj.106_

## Round 0.1 · original submission · Minor Revisions

First of all, congratulations for this excellent work. Haas et al. report on the study of oxygen distribution patterns in corals, algae, and coral-algae interfaces. The novel approach disclosed the existence of hypoxic zones at coral-algae interfaces. This study contributes to our understanding on the possible main biochemical drivers of coral reef phase shift worldwide. Please include a paragraph discussing your findings in the context of the phase shift phenomenon and microbialization observed in different reefs, with the massive increase on benthic algae/turf cover and concomitant coral cover sharp decrease (Bruce et al. 2012; Francini et al., 2013; McDole et al. 2012).

Minor: “ However, planar optodes have never been used to investigate oxygen distribution patterns in the pelagic environment,” I think instead of pelagic the authors meant coral reef systems.

Reviewer 1 ·

Basic reporting

Hass et al. reported on the use of planar optodes for exploring dissolved oxygen patterns on coral algae interactions. Albeit simple, it was a very good idea and application of oxygen optodes, which I really appreciate in knowing more about. The manuscript is well written and organized, as well are the results & discussion.

Experimental design

Despite its enormous potential and applications, oxygen optodes are not very well known to oceanographic community. To overcome this it would be interesting if authors could explain a little more on its functioning. It would be interesting if authors can provide more detail in sensor calibration, even providing some calibration curves. It would give readers more confidence on the data presented. It was mentioned a manuscript in preparation, however more information on this ms will be interesting.


Please consider display oxygen results also in mL.L-1. Despite autors are using the recommended unity, it is seldom used and most oceanographers uses mL-L-1.

Validity of the findings

No Comments

---

## Round 0.2 · accepted · Accept

Congratulations on the study.